# Electrical Stimulation Generates Induced Tumor-Suppressing Cells, Offering a Potential Option for Combatting Breast Cancer and Bone Metastasis

**DOI:** 10.3390/ijms26031030

**Published:** 2025-01-25

**Authors:** Changpeng Cui, Yinzhi Xu, Xue Xiong, Uma K. Aryal, Andy Chen, Stanley Chien, Lidan You, Baiyan Li, Hiroki Yokota

**Affiliations:** 1Department of Pharmacology, School of Pharmacy, Harbin Medical University, Harbin 150081, China; changpengcui5@gmail.com (C.C.); xu2019@purdue.edu (Y.X.); xiong177@purdue.edu (X.X.); 2Weldon School of Biomedical Engineering, Purdue University in Indianapolis, Indianapolis, IN 46202, USA; 3Purdue Proteomics Facility, Bindley Bioscience Center, Purdue University, West Lafayette, IN 47907, USA; uaryal@purdue.edu; 4Department of Comparative Pathobiology, Purdue University, West Lafayette, IN 47907, USA; 5Department of Medical Genetics, Indiana University School of Medicine, Indianapolis, IN 46202, USA; andychen@iu.edu; 6Elmore Family School of Electric and Computer Engineering, Purdue University in Indianapolis, Indianapolis, IN 46202, USA; yschien@purdue.edu; 7Department of Mechanical and Materials Engineering, Queen’s University, Kingston, ON K7L 3N6, Canada; you.lidan@queensu.ca; 8Indiana University Simon Comprehensive Cancer Center, Indianapolis, IN 46202, USA; 9Indiana Center for Musculoskeletal Health, Indianapolis, IN 46202, USA; 10Department of Anatomy, Cell Biology and Physiology, Indiana University School of Medicine, Indianapolis, IN 46202, USA

**Keywords:** iTS cells, bone metastasis, breast cancer, electrical stimulation, tumor-suppressive protein

## Abstract

Treating advanced metastatic cancer, particularly with bone metastasis, remains a significant challenge. In previous studies, induced tumor-suppressing (iTS) cells were successfully generated through genetic, chemical, and mechanical interventions. This study investigates the potential of electrical stimulation to generate iTS cells. Using a custom electrical stimulator with platinum electrodes, mesenchymal stem cells (MSCs) and Jurkat T cells were stimulated under optimized conditions (50 mV/cm, 10–100 Hz, 1 h). Conditioned medium (CM) from electrically stimulated cells demonstrated tumor-suppressing capabilities, inhibiting tumor cell migration, 3D spheroid growth, and cancer tissue fragment viability. Additionally, the CM reduced osteoclast maturation while promoting osteoblast differentiation. Proteomic analysis revealed enrichment of tumor-suppressing proteins, including histone H4, in the CM. Functional studies identified Piezo1 as a key mediator, as its knockdown significantly impaired the tumor-suppressive effects. Mechanistically, the process was distinct from other methods, such as mechanical vibration, with SUN1 inhibition showing no effect on iTS cell generation by electrical stimulation. These findings demonstrate the efficacy of electrical stimulation in enhancing the antitumor capabilities of MSCs and T cells, offering a novel approach to cancer therapy. Further exploration of this strategy could provide valuable insights into developing new treatments for metastatic cancer.

## 1. Introduction

Treating tumor growth in bone, such as metastatic tumors associated with breast cancer and primary bone cancer, presents significant challenges [1]. While antigen-targeted T cell immunotherapy has shown promise in blood cancers, its effectiveness in solid tumors remains limited due to difficulties in antigen selection and T cell infiltration [2]. These limitations highlight the necessity for innovative approaches beyond current paradigms. Enhancing the anti-cancer action of T cells using malignant genetic alterations has been reported [3,4], though there is a safety concern regarding the utilization of tumorigenic mutations. It has been demonstrated that T cells with CARD11-PIK3R3 gene fusion mutations exhibit increased tumor suppression by elevating antigen-dependent CARD-BCL10-MALT1 signaling and NFkB activation.

Similarly, we transformed normal MSCs, tumor cells, and osteoblasts into induced tumor-suppressing (iTS) cells through the overexpression of oncogenes or activation of oncogenic signaling pathways [5,6,7]. By targeting these regulatory pathways, the cells not only experienced enhanced proliferative activities but also acquired potent tumor-suppressive properties. To mitigate potential side effects from oncogene overexpression or oncogenic signaling activation, we developed an alternative mechanical stimulation-based procedure [8]. The rationale behind this biophysical approach is that any process capable of stimulating cell proliferation could potentially catalyze the generation of iTS cells. In this study, we focused on utilizing electrical stimulation (ES) to generate iTS cells.

Electrical fields have diverse applications in biomedical research, from electroporation and cell manipulation to electrophysiology and tissue engineering [9,10,11]. In cell physiology, ES is known to modulate cell growth depending on the current and frequency [12]. The optimal conditions for stimulating cell proliferation may differ among cell types and environmental conditions [13]. Our objective is to leverage ES as a transformative tool to convert MSCs, tumor cells, and Jurkat T cells into iTS cells. ES has been widely used in biomedical research to regulate cellular proliferation and differentiation [12,13,14]. It is reported that ES to cancer cells inhibit their growth [15]. However, its application in generating iTS cells remains unexplored [14]. In this study, we hypothesized that a voltage-gated ion channel, Piezo1, plays a crucial role in mediating MSC responses to ES. Of note, Piezo1 is known to regulate cellular responses to mechanical stimuli [16,17].

Regardless of the method employed to generate iTS cells, the conditioned medium derived from these iTS cells is enriched with numerous tumor-suppressing proteins [18,19,20,21]. Our investigation aimed to determine whether iTS cells generated through ES also secrete these same tumor-suppressing proteins, which were detected in response to the activation of Wnt, PI3K, and PKA signaling [22,23,24]. Additionally, we tested the potential involvement of SUN1, an inner nuclear membrane protein, in responses to ES. SUN1 is a core part of the LINC complex and is required for generating cytoskeletal force [25], but its role in the responses to ES remains elucidated. This study aims to expand our understanding of iTS cell technology and its potential applications in cancer treatment.

## 2. Results

### 2.1. Effect of ES on MSCs and MDA-MB-231 Breast Cancer Cells

A schematic representation of our experimental setup illustrates a pair of platinum wires spaced 2 cm apart, connected to a power supply (Appendix A). We measured the current in response to input voltages of 0.1, 0.2, and 0.5 V across various input frequencies (Hz) (Appendix A). In the MTT-based viability assays, the viability of MSCs and MDA-MB-231 breast cancer cells significantly varied depending on the voltage and frequency of ES. To generate iTS cells, we were interested in the conditions that elevated cell viability. The viability was decreased notably in response to 0.1 V ES (50 mV/cm) at a frequency of 1 kHz (Appendix A), with a smaller reduction observed at lower frequencies. However, MDA-MB-231 cells exhibited increased viability at 10, 50, and 100 Hz. Notably, at 0.5 V, both MSCs and MDA-MB-231 cells showed decreased viability across all frequencies (Appendix A). For two T cell lines (mouse EL4 and human Jurkat), the MTT-based viability was elevated in response to 0.05 and 0.1 V at 10 to 50 Hz (Figure 1A,B). Collectively, we hereafter focused mostly on 0.05–0.1 V at frequencies ranging from 10 to 100 Hz to elevate cell viability for generating iTS cells, although the proper iTS cell-generating conditions may vary depending on the specific cell types involved.

### 2.2. Tumor-Suppressive Effect of ES-Treated MSC CM

We applied 0.1 V (50 mV/cm) to MSCs at frequencies of 5, 10, 50, and 100 Hz. The conditioned medium (CM) from ES-treated MSCs notably reduced the MTT-based viability of various cancer cell lines, including MDA-MB-231 and MDA-MB-436 breast cancer cell lines (Figure 2A), as well as TT2 osteosarcoma (OS) and PANC1 pancreatic ductal adenocarcinoma (PDAC) cells (Figure 2B). Moreover, scratch-based migration assays demonstrated a significant anti-migratory effect of ES-treated MSC CM across frequencies ranging from 5 to 50 Hz (Figure 2C). Western blot analysis revealed an upregulation of p-Akt, a key enzyme in PI3K signaling, in ES-treated MSCs (Figure 2D). To verify the inhibitory effect of ES-treated conditioned medium on the migration ability of tumor cells, we also confirmed through Western blotting the reduction in the levels of EMT-related markers, Snail and Slug, in tumor cells that were cultured in ES-treated CM (Figure 2E). Additionally, the tumor-suppressing capabilities of ES-treated MSC CM were weaker when using ES at a higher voltage of 0.2 V and 0.5 V, compared to 0.1 V across all tested cell lines (MDA-MB-231, MDA-MB-436, TT2, and PANC1) (Appendix A). Regarding the effect of frequencies, the MTT-based viability assay showed that a frequency of 10 Hz presented the most potent anti-cancer effect across the four tested cell lines compared to other higher frequencies (Appendix A).

### 2.3. Tumor-Suppressive Effect of ES-Treated MDA-MB-231 CM

Given that ES was able to generate iTS cells from MSCs, we next subjected MDA-MB-231 breast cancer cells to ES at 0.1 V across frequencies of 5, 10, 50, and 100 Hz. The CM derived from ES-treated MDA-MB-231 cells significantly reduced the MTT-based viability of both MDA-MB-231 and MDA-MB-436 cell lines (Figure 3A), as well as TT2 OS and PANC1 cell lines (Figure 3B). Notably, scratch-based migration assays revealed that ES-treated MDA-MB-231 CM exhibited a significant anti-cancer effect with 0.1 V ES at 10 Hz (Figure 3C). The MTT-based viability assay revealed that 0.1 V ES at 10 Hz presented a potent anti-cancer effect across all tested cell lines (MDA-MB-231, MDA-MB-436, TT2, and PANC1) (Appendix A). This finding represents initial evidence that ES can be used to convert both non-tumor and tumor cells into iTS cells.

### 2.4. Tumor-Suppressive Effect of ES-Treated Jurkat CM

To extend the role of ES in transforming MSCs and MDA-MB-231 cells into iTS cells, we examined the possibility of generating iTS cells from the Jurkat T lymphocyte cell line. In response to ES with 0.1 V at 5, 10, 50, and 100 Hz, the CM from ES-treated Jurkat cells substantially decreased the MTT-based viability of the MDA-MB-231 and MDA-MB-436 cell lines (Figure 4A), as well as TT2 OS cells and PANC1 cells (Appendix A), although the inhibitory effect on TT2 cells was weaker than those on other cancer cell lines.

The study also investigated the impact of combining ES-treated Jurkat CM with Taxol, a commonly used chemotherapeutic agent, on MDA-MB-231 breast cancer cells. The result demonstrated that the combination of ES-treated Jurkat CM with Taxol led to a more pronounced reduction in MTT-based cell viability of MDA-MB-231 cells, by significantly reducing IC_50_ of Taxol from 0.8 mM to 0.2 mM (*p* < 0.05; Figure 4B). Additionally, scratch-based migration assays showed that ES-treated Jurkat CM alone exhibited significant anti-cancer effects, particularly with 0.1 V ES at 10 Hz (Figure 4C). Furthermore, the ES-treated Jurkat CM was found to diminish EdU-based proliferation and transwell-based invasion of MDA-MB-231 cells (Figure 4D,E). Importantly, these effects were consistent with observations made with MSC-derived iTS CM, suggesting a potential avenue for further exploration in cancer therapy.

### 2.5. Bone Protection Effect by ES-Treated Jurkat CM

We continued to examine the effect of ES-treated Jurkat CM on bone homeostasis since breast cancer preferentially metastasizes to bone. Hereafter, we mostly focused on T cell lines and primary T cells because of their common use in cell-based cancer treatment. We demonstrated that the ES-treated Jurkat CM significantly inhibited the differentiation of RANKL-stimulated osteoclasts, resulting in a decreased number of multinucleated TRAP-positive osteoclasts (Figure 5A). Furthermore, MC3T3 osteoblasts, cultured in ES-treated Jurkat CM in 4 weeks, elevated alizarin red staining detecting for mineralized calcium deposits (Figure 5B).

Considering that the inhibitory effects can be influenced by geometric and compositional configurations, we investigated the impact of ES-treated CM using three-dimensional tumor spheroids comprised of MDA-MB-231 breast cancer cells, as well as freshly isolated human breast cancer tissues. The findings demonstrated that ES-treated Jurkat CM significantly reduced the size of the tumor spheroids (*p* < 0.01; Figure 5C), as well as the size of the tissue fragments after 96 h (*p* < 0.05; Figure 5D).

### 2.6. Mass Spectrometry-Based Proteomic Analysis of CM

Having observed the tumor-suppressive capability of CM, we conducted mass spectrometry-based global proteomics and identified a group of proteins enriched in ES-treated Jurkat cell-driven CM. Consistent with the proliferative nature of iTS cells, protein enrichment analysis revealed that K-Ras signaling is enriched among the ES-treated iTS proteomes (Figure 6A). Furthermore, the comparison of enriched proteins in the CM with those identified from five independent procedures for generating iTS cells revealed that many proteins were elevated by the other iTS generation methods. For instance, nine proteins—Hsp90ab1, Flna, PKM, Hsp8, Eno1, MSN, Eef2, Ppia, and Pgam1—were enriched in the six differently generated CM, including the activation of Wnt signaling, PI3K signaling, and PKA signaling, the overexpression of Oct4, the application of low-intensity vibration (LIV), and the application of ES (Figure 6B). Additionally, seven proteins (Flnb, Myh9, Vcp, Hspa4, Ywhae, Got1, and Prdx1) and five proteins (Actg1, Eef1a1, Tuba1b, Aldoa, and Arhgdia) were elevated by the five independent iTS generating procedures, including ES treatment (Figure 6B).

Our data also revealed a unique proteomic signature associated with ES treatment. The volcano plot shows these uniquely enriched proteins, including Histone H4 (Figure 6C). Previous research suggests that extracellular Histone H4 can act as a tumor suppressor by interacting with a toll-like receptor [5,18]. Western blot analysis confirmed that ES treatment elevated the levels of Histone H4, Hspa8, MSN, Eef2, Aldoa, and Eno1, which is consistent with previous studies demonstrating Eno1’s tumor-suppressive function [19,26].

### 2.7. Mechanism of Piezo1 in ES-Driven iTS Cell Generation

To investigate the factors influencing MSCs’ response to ES and their transformation into iTS cells, we hypothesized that Piezo1 plays a central role, alongside potential regulatory effects of SUN1 and Wnt signaling proteins. Piezo1 was knocked down in MSCs using siRNA (Figure 7A), followed by ES treatment and CM collection. MTT assays showed that Piezo1 knockdown significantly reduced the inhibitory effect of ES-generated CM on breast cancer cell viability (Figure 7B), indicating Piezo1’s essential role in MSC response to ES and tumor-suppressive CM production. Scratch assays confirmed that CM from Piezo1-knockdown MSCs failed to inhibit MDA-MB-231 cell migration (Figure 7C).

SUN1 overexpression hindered MSC transformation into iTS cells under ES without affecting K-Ras and c-Myc levels (Appendix A). In contrast, SUN1 knockdown enhanced CM’s tumor-suppressive effects, reducing viability and migration of MDA-MB-231 and MDA-MB-436 cells (Appendix A–F), suggesting that SUN1 acts as a negative regulator. Although overexpression of Wnt proteins LRP5 and β-catenin alone could transform MSCs into iTS cells with tumor-suppressive CM (Appendix A), co-treatment did not enhance ES effects (Appendix A). This highlights Piezo1 as the primary mediator of ES-driven tumor suppression. In conclusion, Piezo1 is central to MSC response to ES, while SUN1 negatively regulates this process.

### 2.8. Variations in Sensitivity to ES-Treated T-Cell-Derived CM

To further investigate the conversion of T cells into iTS cells, we used the murine T cell line EL4. After exposing EL4 cells to ES at 0.05 or 0.1 V and frequencies of 10 and 50 Hz for 1 and 2 h, EL4-derived CM reduced the viability of MDA-MB-436 breast cancer cells in MTT assays. However, no significant effect was observed on MDA-MB-231 cells, indicating varying sensitivities to T-cell-derived CM among cancer cell lines (Appendix A). Western blot analysis following ES showed increased levels of p-Akt and c-Myc in EL4 cells, indicating the activation of proliferative signaling (Appendix A).

## 3. Discussion

This study introduces a novel method for generating iTS cells and inhibiting the maturation of bone-resorptive osteoclasts through ES treatment, marking the first instance of such a technique (Figure 8). iTS cells were successfully derived from MSCs, T cells, and cancer cells, and their CM effectively suppressed tumor progression and prevented the formation of the osteolytic environment. The applied electrical conditions were 50 mV/cm at frequencies ranging from 1 to 100 Hz, with an approximate current of 0.1 mA. Notably, due to the capacitance of both the medium and the cells, the current is dependent on the input frequency. Specifically, as the frequency increases, the current amplifies and becomes less responsive to changes in voltage input. The findings indicate that an electrical field strength of 50 mV/cm is appropriate for the generation of iTS cells using MSCs and T cells as host cells.

The selection of cancer cell lines was based on their aggressive phenotypes and clinical relevance. Specifically, we utilized MDA-MB-231 and MDA-MB-436, both of which represent triple-negative breast cancer subtypes, characterized by poor prognosis and limited treatment options. Additionally, osteosarcoma (TT2) and pancreatic ductal adenocarcinoma (PANC1) cell lines were included to evaluate the generalizability of ES-induced tumor-suppressive effects beyond breast cancer cells. However, the absence of estrogen receptor-positive (ER+) breast cancer cell lines or other TNBC subtypes limits the broader applicability of the findings to diverse breast cancer phenotypes.

We have validated the tumor-suppressive potential of CM across diverse cancer cell lines, cancer spheroids, and in combination with the chemotherapy drug Taxol. Additionally, our findings indicate CM’s efficacy in inhibiting the maturation of bone-resorbing osteoclasts and promoting the differentiation of bone-forming osteoblasts. These attributes align with those observed in other CM derived from iTS cells, which were produced through mechanical stimulation, oncogene overexpression, and activation of tumorigenic pathways, like Wnt, PI3K, and PKA. Notably, the impact of ES varies significantly depending on the frequency-linked current: while low currents are stimulatory, high currents exert inhibitory effects. This mirrors the effects of mechanical stimulation, where low intensity is stimulatory while high intensity is detrimental.

The tumor-suppressive action of iTS cells does not require direct interactions with tumor cells, as iTS cells secrete a group of proteins that function as tumor suppressors in the extracellular domain. We have previously demonstrated that nine proteins (EEF2, PKM, PPIA, ENO1, PGAM1, MSN, FLNA, HSPA8, HSP90AB1) were enriched in iTS CM produced by five independent iTSC-generating procedures [27]. These procedures include the activation of Wnt, PI3K, and PKA pathways, the overexpression of Oct4, and the application of mechanical vibration. In the current study, we have shown that these nine proteins were also enriched in ES-treated Jurkat cell-derived CM. Furthermore, ES treatment also elevated the level of Histone H4 which is known to act as a tumor suppressor in the extracellular domain [28].

This study shows that ES activates Akt phosphorylation (p-Akt) in MSCs, contributing to the generation of iTS cells. Although the exact mechanism is unclear, PI3K signaling is involved, as shown in our prior work [20]. ES may alter chromatin structure, influencing transcriptional pathways like PI3K signaling. Additionally, Piezo1 and YAP1, key in mechanical stimulation, may also contribute to Akt activation in ES. Future research should explore the relationship between ES, chromatin remodeling, and mechano-transduction.

Piezo1 is crucial for mechano-transduction, converting physical stimuli into biochemical signals that regulate proliferation, differentiation, and migration. In this study, siRNA-mediated Piezo1 knockdown in MSCs impaired their response to ES, reducing tumor-suppressive CM production. This highlights Piezo1’s role in ES responses and its interaction with pathways like PI3K/Akt and calcium signaling, which are critical for tumor-suppressive secretome generation. While Piezo1 facilitates the anti-cancer effects of ES, SUN1 acts as a negative regulator. SUN1 overexpression inhibited iTS cell generation, whereas its knockdown enhanced tumor-suppressive activity. This underscores distinct roles within the mechano-transduction machinery: Piezo1 drives positive responses to ES, while SUN1 may regulate or limit excessive responses. In addition to Piezo1, other voltage-gated ion channels are likely to contribute to ES-induced cellular effects by regulating ion flow (e.g., calcium, potassium, sodium). For example, calcium influx through voltage-gated calcium channels may interact with Piezo1 signaling to modulate responses. While the exact roles of these channels are not fully understood, their involvement could explain the diverse effects of ES.

Super competition is a phenomenon in cancer where oncogene-activated cells gain an unfair advantage over their healthy neighbors [29,30,31]. These “super-competitors” grow much more quickly due to the oncogene’s influence, and oncogenes often make cells less susceptible to signals that normally trigger cell death. While super competition is not directly linked to generating super-competitors through specific procedures, like ES treatment, the principle of selective elimination of tumor cells by super-competitors holds promise. Previous studies suggest that CM from iTS cells can selectively eliminate tumor cells [7,20,26]. This selectivity is an important area for further investigation, and understanding how these cells achieve this selectivity could lead to the development of novel cancer therapies.

ES treatment is a non-invasive procedure that is predicted to lessen side effects while improving the efficacy of anti-cancer treatments. However, considerable experimental and clinical research is needed to precisely manage parameters such as the strength, frequency, and stimulation period of ES to achieve optimal therapeutic results with iTS cells and their CM. The application of ES in clinical settings is not entirely unprecedented. Optune [32], an FDA-approved device, utilizes alternating electric fields to treat glioblastomas, demonstrating the feasibility of ES-based therapies in clinical practice. This serves as a foundation for exploring the scalability and regulatory pathway for ES devices targeting other cancers. However, adapting this technology for broader use presents challenges, such as designing scalable devices capable of delivering precise electrical parameters to different tissues. Additionally, ensuring safety and efficacy across diverse cancer types requires rigorous preclinical and clinical evaluations. The safety of long-term ES on normal cells and tissues is not fully understood, and potential side effects require further investigation. The release and mode of action of tumor-promoting proteins are complex, and developing methods to ensure their stability and efficiency in vivo, while preventing escape or resistance by tumor cells, is a critical area of study.

## 4. Materials and Methods

### 4.1. Electronic Stimulator

A bespoke electronic stimulator was crafted using a pair of platinum wires with a diameter of 240 µm. These wires were positioned in parallel with a 2 cm separation and submerged in a culture dish filled with a culture medium. The stimulator is characterized by a capacitance of 2.63 × 10^−6^ F in series with a resistance of 116 Ω with cells (144 Ω without cells). An input voltage of a sinusoidal waveform, with an amplitude of 100 mV, was applied to the wires across the frequency range of 1 to 10,000 Hz for 1 h. The resulting current measurements were 0.012 mA (at 10 Hz), 0.057 mA (at 50 Hz), and 0.11 mA (at 100 Hz). Notably, the temperature elevation of the culture medium during the 1 h-stimulation period was estimated to be less than 0.01 °C.

### 4.2. Cell Culture

Six different tumor and bone cell types were cultured in DMEM (Corning Inc., Corning, NY, USA): two human triple-negative breast tumor cell lines, MDA-MB-231 and MDA-MB-436 (ATCC, Manassas, VA, USA), PANC1 pancreatic cancer cells (ATCC), xenograft TT2-77 osteosarcoma cells, RAW264.7 pre-osteoclast cells (ATCC), and murine MSCs derived from the bone marrow of C57BL/6 mice. Jurkat lymphocytes were cultured in RPMI-1640 (Gibco, Carlsbad, CA, USA), and MC3T3-E1 osteoblasts (Sigma, St. Louis, MO, USA) were grown in αMEM (Gibco). Tumor cells were treated with the chemotherapeutic agent Taxol (3257, Tocris Bioscience, Bristol, UK). The culture medium was supplemented with 10% fetal bovine serum and antibiotics (100 units/mL penicillin, and 100 µg/mL streptomycin; Life Technologies, Grand Island, NY, USA).

### 4.3. MTT and EdU Assays

Using 96-well plates with approximately 2000 cells seeded in each well, an MTT-based metabolic activity assay was conducted. Cells were incubated for two days with treatment agents, and on day 4 they were dyed with 0.5 mg/mL thiazolyl blue tetrazolium bromide (M5655, Sigma). Optical density for assessing metabolic activities was determined at 562 nm. EdU-based proliferation activity was evaluated using 96-well plates with approximately 1000 cells in each well. A fluorescence-based cell proliferation kit (Click-iT™, EdU Alexa Fluor™ 488 Imaging Kit; Thermo Fisher, Waltham, MA, USA) was employed, following the procedure described by the manufacturer.

### 4.4. Scratch Assay and Transwell Invasion Assay

To evaluate the two-dimensional motility of cancer cells, a wound-healing scratch assay was utilized. On day 1, approximately 3 × 10^5^ cells were seeded in 12-well plates, and on day 2 a scratch was made on the cell layer with the tip of a plastic pipette. Using a light microscope (40×), cell-free zones were imaged at 0 h and 24 h, and the alteration in their areas was quantified using ImageJ. In response to conditioned medium (CM), the invasion capacity of cancer cells was determined using 8-µm pore transwell chambers (353182, Thermo Fisher) in a 12-well plate. The chambers were coated with 300 µL Matrigel (100 µg/mL), and 500 µL of the serum-free medium was added. The chamber was washed three times with the serum-free medium. Approximately 7 × 10^4^ cells in 300 µL serum-free DMEM were placed in the upper chamber, and 700 µL medium was added to the lower chamber. Cells that moved to the lower side of the membrane were fixed and stained with methanol and crystal violet. Five randomly chosen images were taken with an inverted optical microscope (100×), and the average number of stained cells, representing the invasion capacity, was determined.

### 4.5. Western Blot Analysis

Cells were lysed with RIPA lysis buffer (sc-24948, Santa Cruz Biotech, Dallas, TX, USA) containing protease inhibitors (PIA32963, Thermo Fisher) and phosphatase inhibitors (2006643, Calbiochem, Billerica, MA, USA). Proteins were separated by 10–15% SDS gels (Bio-Rad Laboratories, Hercules, CA, USA) and transferred to a polyvinylidene difluoride membrane (IPVH00010, Millipore, Billerica, MA, USA). The membrane was incubated with primary antibodies, followed by incubation with secondary antibodies (7074S/7076S, Cell Signaling, Danvers, MA, USA). Antibodies used included those against Snail, Slug, p-Src, Src, p-Akt, Akt, c-Myc, LRP5, ALDOA, Eno1, β-catenin (Cell Signaling), K-Ras (Santa Cruz Biotech), Piezo1, and SUN1 (Proteintech, Rosemont, IL, USA), with β-actin (Sigma) as a control. Protein levels were determined using a SuperSignal West Femto maximum sensitivity substrate (PI34096, Thermo Fisher) and a luminescent image analyzer (LAS-3000, Fuji Film, Tokyo, Japan).

### 4.6. D Spheroid Assay

MDA-MB-231 tumor cells were seeded at approximately 5000 per well in a U-shaped 96-well plate (100 μL per well) (S-Bio, Hudson, NH, USA). Cells were cultured for 24 h to form a spheroid. Photos and records were taken every 24 h, and changes in cell morphology were continuously recorded over 96 h.

### 4.7. Ex Vivo Assays

Human breast and prostate cancer tissues were obtained from the Simon Cancer Center Tissue Procurement Core with approval from the Indiana University Institutional Review Board (#1911155674). Each tissue sample (~1 g) was manually fragmented into small pieces (0.5~0.8 mm in length) using a scalpel. These fragments were cultured in DMEM supplemented with 10% fetal bovine serum and antibiotics for one day. Subsequently, CM was added for four more days, and changes in the size of at least four fragments were monitored and recorded.

### 4.8. Plasmid Transfection and RNA Interference

The overexpression of SUN1 was accomplished with plasmids (#125851, Addgene, Cambridge, MA, USA), while blank plasmids (FLAG-HA-pcDNA3.1; Addgene) served as the control. Plasmids were transfected into 70–90% confluent cells using Lipofectamine 3000 (#L300015, Thermo Fisher). RNA interference was conducted using siRNA specific to SUN1 (#AM16708; Ambion, Austin, TX, USA) and Piezo1 (#4392420; Ambion) with a negative siRNA (Silencer Select #1, Thermo Fisher) as a nonspecific control, following the previously described procedure.

### 4.9. Mass Spectrometry-Based Proteomics Analysis

#### 4.9.1. Proteomics Sample Preparation

Proteomics analysis was performed as described earlier [33,34]. Briefly, media samples were buffer exchanged to remove contaminants in the growth medium, and then proteins were precipitated using four volumes of cold (−20 °C) acetone. The precipitated proteins were collected by centrifugation at 17,200× *g* for 20 min at 4 °C, washed 3× with 80% cold (−20 °C) acetone, and processed for MS analysis. The protein pellets were dissolved in 8M urea by incubating for 1 h at room temperature with continuous vortexing and protein concentration was measured using the bicinchoninic acid (BCA) assay (Pierce Chemical Co., Rockford, IL, USA), using bovine serum albumin as a standard. Fifty (50) µg of total protein (or equivalent volume) was reduced by incubating with dithiothreitol (DTT) at 10 mM final concentration for 45 min at 55 °C. The samples were then cysteine alkylated with iodoacetamide to a final concentration of 20 mM and incubated in the dark at room temperature for 45 min. The samples were again treated with DTT to a final concentration of 5 mM at room temperature for 30 min to scavenge residual IAA. The reduced and alkylated samples were then diluted by adding 25 mM ammonium bicarbonate (ABC) to lower urea concentration below 2 mM. The samples were digested with trypsin overnight at 37 °C using a protein/trypsin ratio of 50:1. Following overnight digestion, peptides were desalted using the Mini Pierce Peptide Desalting Spin Columns following the protocol provided by the manufacturer (Thermo Fisher). Dried desalted peptides were resuspended in a resuspension solution composed of 3% ACN and 0.1% FA in water to a final concentration of 0.2 μg/μL and 1 μg of peptides (5 μL) were used for subsequent LC−MS/MS analysis as described below.

#### 4.9.2. LC-MS/MS Data Acquisition

The reconstituted peptides were analyzed by reverse-phase separation using a Dionex UltiMate 3000 RSLC system coupled with Orbitrap Fusion Lumos mass spectrometer (Thermo Fisher). Peptides were first loaded into the trap column (300 µm ID × 5 mm) packed with 5 µm 100 Å PepMap C18 medium (Thermo Fisher) and then separated using a reverse phase 1.7 μm IonOptics Aurora Ultimate C18 column (75 μm × 25 cm). The column was maintained at 40 °C; mobile phase solvent A was 0.1% FA in water and solvent B was 0.1% FA in 80% ACN. The loading buffer was 0.1% FA in 2% ACN. Peptides were loaded into the trap column for 5 min at 5 μL/min and then separated with a flow rate of 300 nL/min using a 130 min linear gradient. The concentration of mobile phase B was increased linearly to 8% in 5 min, 27% B in 80 min, and then 45% B at 100 min. After 100 min, it was subsequentially increased to 100% B at 105 min and held constant for another 7 min before reverting to 2% B in 112.1 min and maintaining at 2% B until the end of the run. The Lc-MS/MS analyses were performed in the Orbitrap Fusion Lumos Tribrid Mass Spectrometer with the orbitrap detector, with a MS1 resolution of 120,000 and MS2 resolution at 7500. Quadrupole isolation was set to “True”. The scan range used was between 375 and 1500 *m*/*z*. RF lenses was set to 30%, AGC target was set to “Standard”, with a Maximum injection time of 50 s and 1 micro-scan. A dynamic exclusion duration of 60 s was used, with exclusion of isotopes. Data-dependent mode “Cycle Time,” with 3 s between scans, was used. HCD (Higher Energy Collisional Dissociation) was used for fragmentation, with HCD collision energy set to 30%. We also used an MS/MS Maximum injection time of 50 s and 1 micro-scan.

#### 4.9.3. LC-MS/MS Data Analysis

Raw LC-MS/MS data were searched using MaxQuant [35] for protein identification and label-free quantitation. Raw spectra were searched against the human uniport protein sequence database (downloaded in July 2024) containing 75,000 protein sequences, including isoforms for protein identification and MS1-based label-free quantitation (LFQ). The minimum length of the peptides was set at 6 AA residues in the database search. The following parameters were edited for the searches: precursor mass tolerance was set at 10 ppm, MS/MS mass tolerance was set at 20 ppm, enzyme specificity of trypsin allowing up to 2 missed cleavages, oxidation of methionine (M) as a variable modification and carba-mido-methylation of cysteine as a fixed modification. The false discovery rate (FDR) of peptide spectral match and protein identification was set to 0.01. The unique plus razor peptides (nonredundant and nonunique peptides assigned to the protein group with most other peptides) were used for peptide quantitation. Only proteins detected with at least one unique peptide and MS/MS (spectral counts) ≥2 were considered valid identifications. LFQ (Label Free Quantitation) intensity values were used for relative protein abundance. LFQ intensity values are typically normalized peptide/protein intensities adjusting raw intensity to account for differences in sample loading, instrument performance, and other technical variations, as well as the size of the proteins. LFQ intensities of peptides are calculated using normalized peptide intensities and the normalization process ensures that the observed differences in protein intensity reflect true biological variations rather than technical artifacts. MaxQuant search results were analyzed using Perseus bioinformatic software [36] for data filtering, visualization, and statistical analysis. Two groups of MSC-derived conditioned media (CM) were control CM and ES-treated CM, and each consisted of three samples. The relative abundance was compared in the two groups using signal intensities as an indicator of protein levels. Differential protein abundance was calculated using a linear model (lm function) in R (v4.3.1). Gene set enrichment analysis was conducted using the fgsea (v1.28.0) package and by evaluating a gene list ranked by -log10 (*p*-value) of each gene signed with the direction of effect of ES. The Hallmark gene set was used.

### 4.10. Statistical Analysis

For cell-based experiments, three or four independent experiments were conducted, and data were expressed as mean ± S.D. Statistical significance was evaluated using a one-way analysis of variance (ANOVA). Post hoc statistical comparisons with control groups were performed using Bonferroni correction, with statistical significance set at *p* < 0.05. The asterisks indicate statistical significance compared to the control group. The single asterisk (*) indicates *p* < 0.05, the double asterisks (**) indicate *p* < 0.01, and the triple asterisks (***) indicate *p* < 0.001. Connecting lines indicate group comparisons. The single sign (#) indicates *p* < 0.05, and double signs (##) indicate *p* < 0.01.

## 5. Conclusions

This study highlights the potential of ES to generate iTS cells and the critical role of Piezo1 in MSC responses to ES. Piezo1, a key mechanosensitive ion channel, facilitates the production of tumor-suppressive CM, with its knockdown significantly impairing anti-cancer effects. Conversely, SUN1 acts as a negative regulator, limiting iTS cell generation. These findings underscore the interplay of mechanosensitive pathways in ES-driven tumor suppression and support targeting Piezo1 to enhance ES-based cancer therapies. As a non-invasive approach, ES shows promise for treating solid tumors and bone metastases, warranting further research into Piezo1-linked mechanisms and clinical applications.

## Figures and Tables

**Figure 1 ijms-26-01030-f001:**
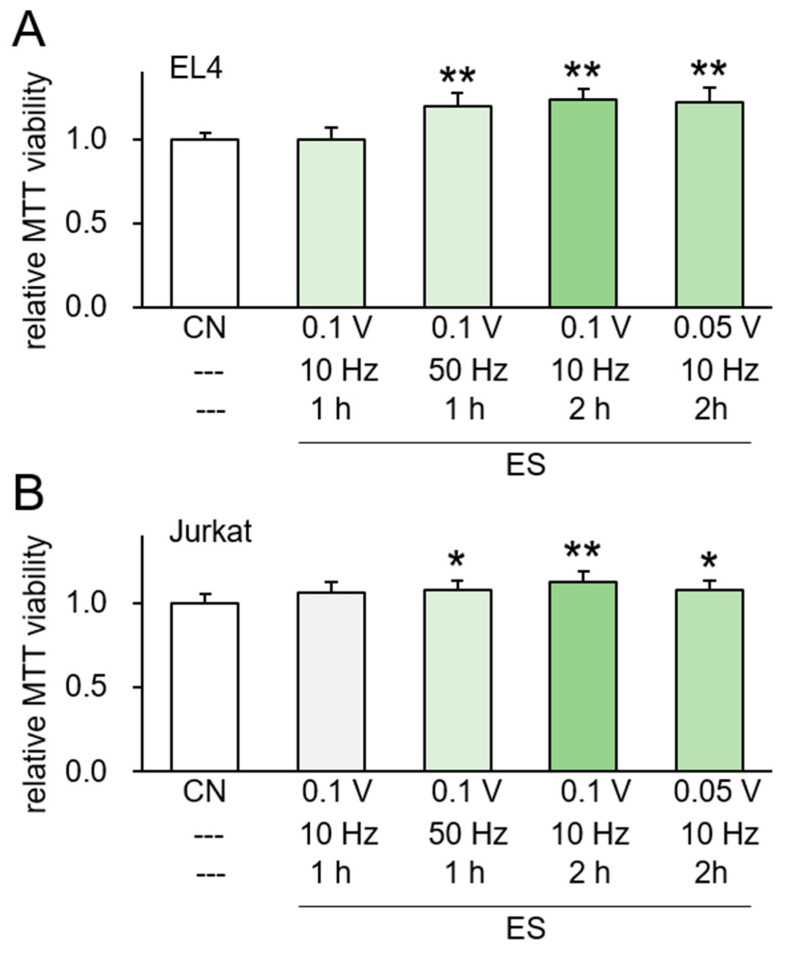
ES effects for EL4 and Jurkat cells directly. CN = control, and CM = conditioned medium. Data are presented as mean ± SD (n = 6 independent experiments). Statistical analysis was performed using one-way ANOVA followed by Bonferroni correction. The asterisks indicate statistical significance compared to the control group. The single asterisk (*) indicates *p* < 0.05, and the double asterisks (**) indicate *p* < 0.01. ES was applied with 0.1 or 0.05 V at 10 or 50 Hz for 1 or 2 h, and the CM was harvested 24 h later. (**A**,**B**) MTT-based viability of EL4 and Jurkat cells in response to ES-treated.

**Figure 2 ijms-26-01030-f002:**
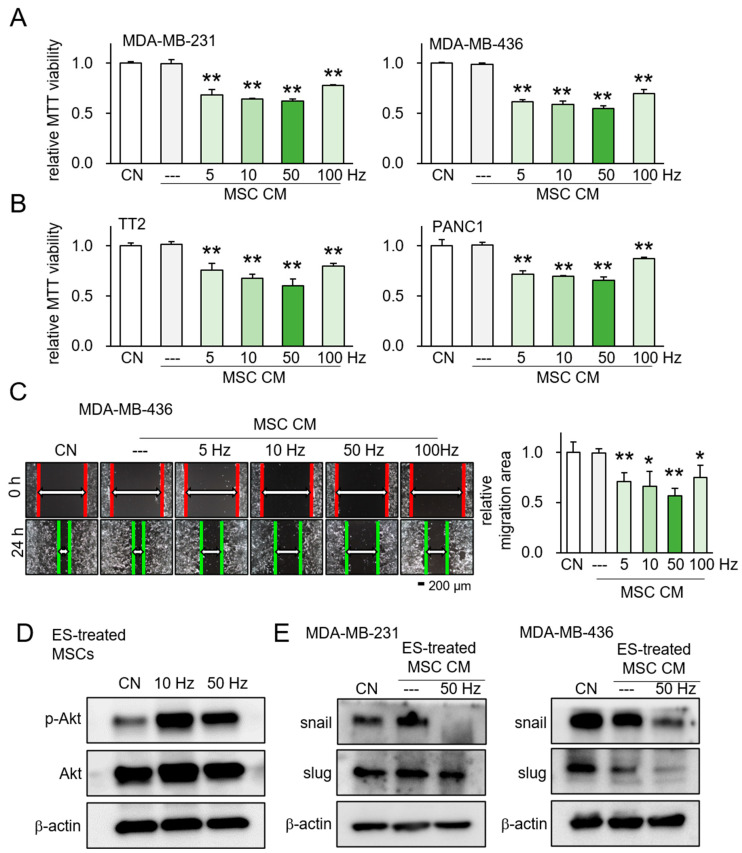
Evaluation of ES-treated MSC CM. CN = control, and CM = conditioned medium. Data are presented as mean ± SD (n = 6 independent experiments). Statistical analysis was performed using one-way ANOVA followed by Bonferroni correction. The asterisks indicate statistical significance compared to the control group. The single asterisk (*) indicates *p* < 0.05, and the double asterisks (**) indicate *p* < 0.01. ES was applied with 0.1 V at 5, 10, 50, and 100 Hz for 1 h, and the CM was harvested 24 h later. (**A**) Reduction in MTT-based viability of MDA-MB-231 and MDA-MB-436 breast cancer cells in response to ES-treated MSC CM. (**B**) Reduction in MTT-based viability of TT2 osteosarcoma cells and PANC1 pancreatic cancer cells in response to ES-treated MSC CM. (**C**) Suppression of scratch-based motility of MDA-MB-436 cells in response to ES-treated MSC CM. (**D**) Elevation in the level of p-Akt in ES-treated MSCs at 10 and 50 Hz for 1 h. (**E**) Decreased level of the EMT-related markers, Snail and Slug, in the two breast cancer cell lines (MDA-MB-231, MDA-MB-436), which were cultured in ES-treated MSC CM.

**Figure 3 ijms-26-01030-f003:**
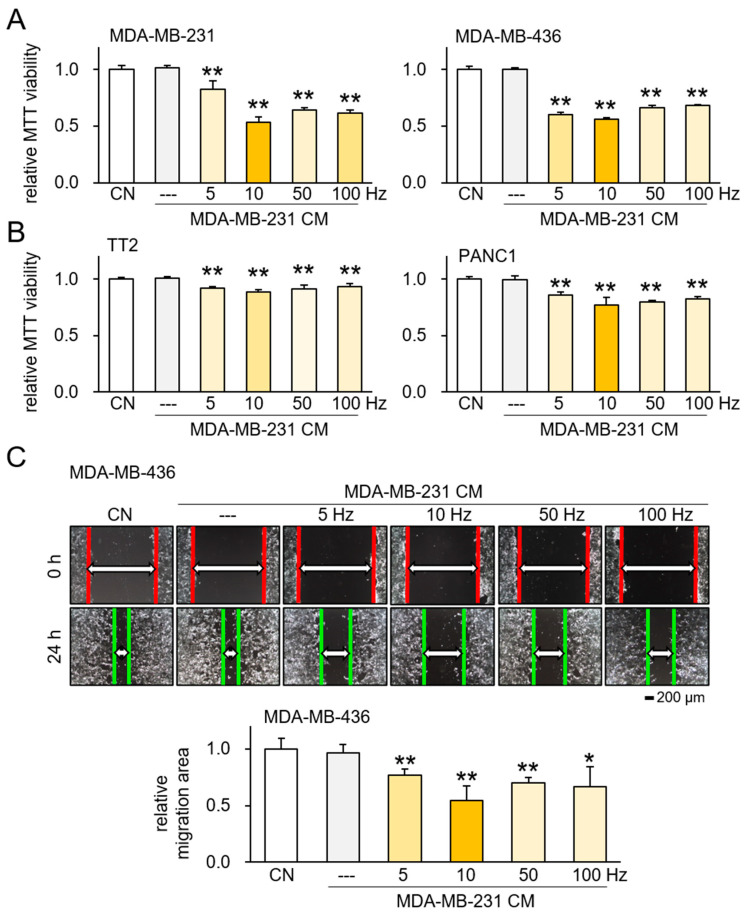
Tumor-suppressive effects of ES-treated MDA-MB-436 CM. ES = electrical stimulation, CN = control, and CM = conditioned medium. Data are presented as mean ± SD (n = 6 independent experiments). Statistical analysis was performed using one-way ANOVA followed by Bonferroni correction. The asterisks indicate statistical significance compared to the control group. The single asterisk (*) indicates *p* < 0.05, and the double asterisks (**) indicate *p* < 0.01. ES was applied with 0.1 V at 5, 10, 50, and 100 Hz for 1 h, and the CM was harvested 24 h later. (**A**) Reduction in MTT-based viability of MDA-MB-231 and MDA-MB-436 cells by ES-treated MDA-MB-231 CM. (**B**) Reduction in MTT-based viability of TT2 and PANC1 cells by ES-treated MDA-MB-231 CM. (**C**) Reduction in scratch-based motility of MDA-MB-436 cells by ES-treated MDA-MB-231 CM.

**Figure 4 ijms-26-01030-f004:**
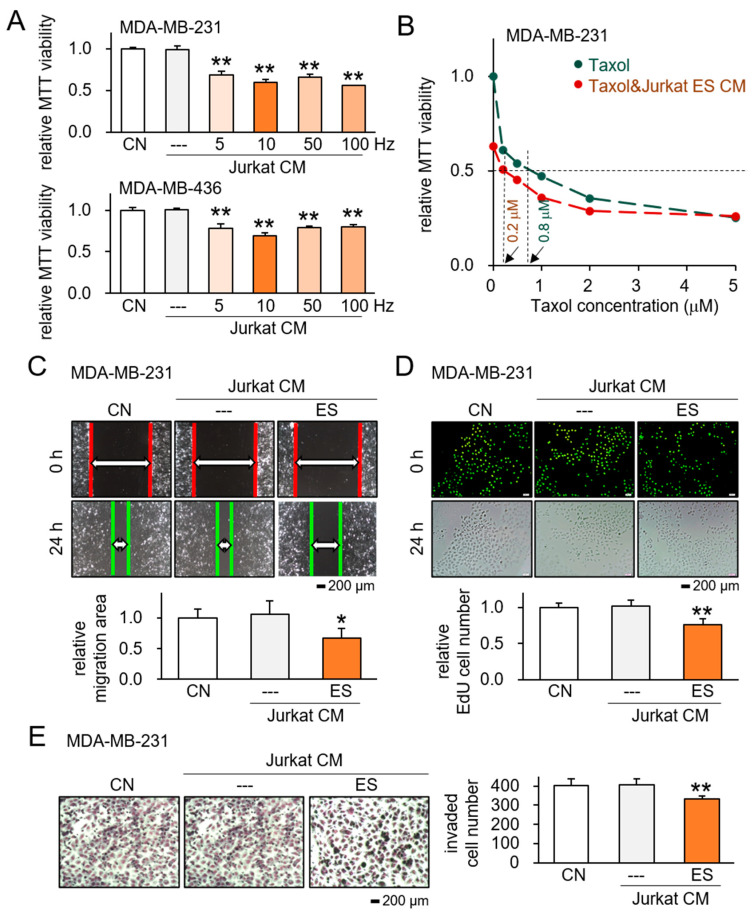
Tumor-suppressive action of ES-treated Jurkat lymphocyte-derived CM. CN = control, and CM = conditioned medium. Data are presented as mean ± SD (n = 6 independent experiments). Statistical analysis was performed using one-way ANOVA followed by Bonferroni correction. The asterisks indicate statistical significance compared to the control group. The single asterisk (*) indicates *p* < 0.05, and the double asterisks (**) indicate *p* < 0.01. ES was applied with 0.1 V at 5, 10, 50, and 100 Hz for 1 h, and the CM was harvested 24 h later. (**A**) Reduction in MTT-based viability of MDA-MB-231 and MDA-MB-436 breast cancer cells by ES-treated Jurkat CM. (**B**) Additive anti-tumor effects of ES-treated Jurkat derived CM with Taxol. (**C**) Suppression of scratch-based motility of MDA-MB-231 cells in response to ES-treated Jurkat CM. (**D**,**E**) Inhibitory effects of ES-treated Jurkat CM on EdU-based proliferation and transwell invasion of MDA-MB-231 cells.

**Figure 5 ijms-26-01030-f005:**
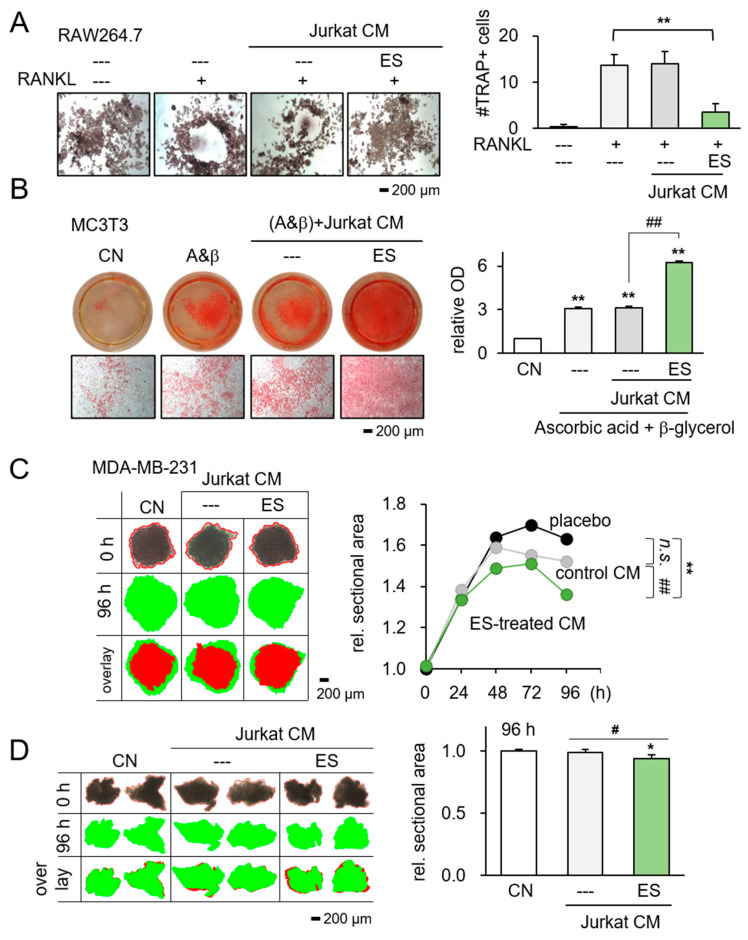
Inhibitory effects of ES-treated Jurkat lymphocyte-derived CM on RAW264.7 osteoclast and breast cancer tissue fragment. CN = control, and CM = conditioned medium, n.s. = nonsense. Data are presented as mean ± SD (n = 6 independent experiments). Statistical analysis was performed using one-way ANOVA followed by Bonferroni correction. The asterisks indicate statistical significance compared to the control group. The single asterisk (*) indicates *p* < 0.05, and the double asterisks (**) indicate *p* < 0.01. Using connecting lines to indicate group comparisons. The single pound sign (#) indicates *p* < 0.05, and the double pound signs (##) indicate *p* < 0.01. ES was applied with 0.1 V at 10 Hz for 1 h, and the CM was harvested 24 h later. (**A**) Significant reduction in multi-nucleated RANKL-stimulated osteoclasts in response to ES-treated Jurkat-derived CM. (**B**) Elevation in Alizarin Red staining of MC3T3 osteoblasts by the administration of ES-treated Jurkat CM. (**C**) 3D tumor spheroids in response to ES-treated Jurkat CM and every 24 h observations and records. (**D**) Reduction in breast cancer tissue fragment size in 96 h in response to ES-treated Jurkat-derived CM (n = 6). The red image indicates tissue fragments at 0 h, while the green image indicates them at 96 h.

**Figure 6 ijms-26-01030-f006:**
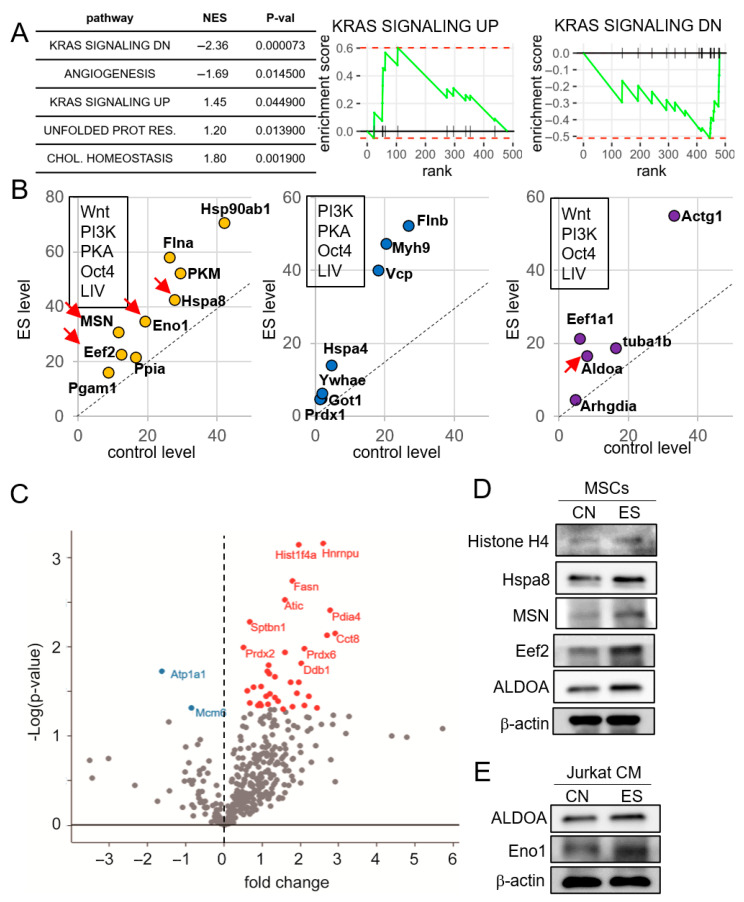
Mass spectrometry-based proteomics analysis of the ES-treated CM. CN = control, ES = electrical stimulation, and CM = conditioned medium. ES was applied with 0.1 V at 10, Hz for 1 h, and the CM was harvested 24 h later from Jurkat cells and MSCs. For the proteins directed by the red arrow, their elevated level by ES treatment was validated by Western blot analysis. (**A**) Enrichment of K-Ras signaling in ES-treated Jurkart cell-derived CM. The data shows the pathways with *p* < 0.05 in gene set enrichment analysis. Enrichment plots of gene sets of upregulation by K-Ras activation (KRAS SIGNALING UP) and downregulation by K-Ras activation (KRAS SIGNALING DN) are shown. (**B**) Nine enriched proteins (Hsp90ab1, Flna, PKM, Hsp8, Eno1, MSN, Eef2, Ppia, and Pgam1) by 6 iTS generation procedures (the activation of Wnt signaling, PI3K signaling, and PKA signaling, the overexpression of Oct4, the application of low-intensity vibration—LIV, and the application of ES), seven proteins (Flnb, Myh9, Vcp, Hspa4, Ywhae, Got1, and Prdx1) and five proteins (Actg1, Eef1a1, Tuba1b, Aldoa, and Arhgdia) by the five independent iTS generating procedures, including ES treatment. (**C**) Volcano plot, showing the proteins enriched by ES treatment, including Histone H4. (**D**) Western blot images, showing the elevated levels of Histone H4, Hspa8, MSN, Eef2, and Aldoa in ES-treated MSC-derived CM. (**E**) Western blot images, showing the elevated levels of Aldoa and Eno1 in ES-treated Jurkat cell-derived CM.

**Figure 7 ijms-26-01030-f007:**
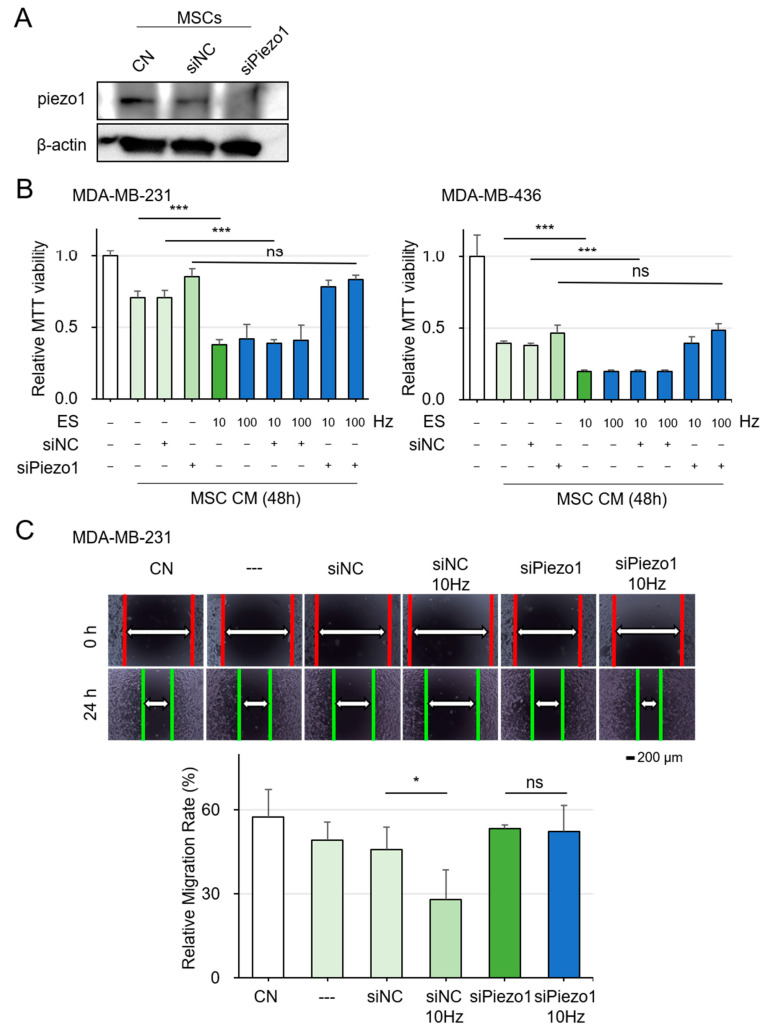
Mechanism of Piezo1 and Wnt Signaling in ES-driven iTS Cell Generation CN = control, and CM = conditioned medium, ns = nonsense. Data are presented as mean ± SD (n = 6 independent experiments). Statistical analysis was performed using one-way ANOVA followed by Bonferroni correction. Using connecting lines to indicate group comparisons. The single asterisk (*) indicates *p* < 0.05, and the triple asterisks (***) indicate *p* < 0.001. ES was applied with 0.1 V at 5, 10, 50, and 100 Hz for 1 h, and the CM was harvested 24 h later. (**A**) Decreased levels of Piezo1 proteins in MSCs by the transfection of si-Piezo1 plasmids. (**B**) Reduction in MTT-based viability of MDA-MB-231 and MDA-MB-436 breast cancer cells in response to ES-treated MSC CM. While knockdown of Piezo1 in MSCs significantly weakened the inhibitory effect of ES-generated conditioned medium (CM) on the viability of breast cancer cells MDA-MB-231 and MDA-MB-436. (**C**) Suppression of scratch-based motility of MDA-MB-436 cells in response to ES-treated MSC CM.

**Figure 8 ijms-26-01030-f008:**
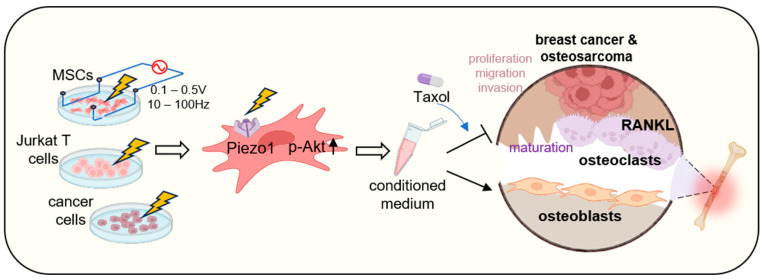
Proposed mechanism of the anti-tumor action of ES effect. The application of ES to MSCs, T cells, and tumor cells triggers the activation of the mechanosensitive ion channel Piezo1 and PI3K/Akt signaling, which plays a pivotal role in cellular responses to ES. The activation leads to the secretion of tumor-suppressive proteins, including Histone H4, enriched in the CM. Together with Taxol, the CM exhibits anti-tumor effects on breast cancer cells and osteosarcoma cells by suppressing cancer cell migration, proliferation, and invasion while inhibiting RANKL-stimulated osteoclast differentiation and enhancing osteoblast mineralization, thereby counteracting bone metastasis.

## Data Availability

The data presented in this study are available in this article (and Appendix A).

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
