# Peer review of "Electrical Stimulation Generates Induced Tumor-Suppressing Cells, Offering a Potential Option for Combatting Breast Cancer and Bone Metastasis"

_ijms, 2025, doi:10.3390/ijms26031030_

Round 1
Reviewer 1 Report
Comments and Suggestions for Authors
Changpeng Cui et al. induced the generation of iTS cells by electrical stimulation, and the characterisation of these cells. Overall, the experiments are well done, but the paper also needs some corrections to the figures and statistics, as well as Material and Methods.
There are a great variety of breast cancer subtypes and a large number of cell lines. It is important to consider whether the results of this experiment should be regarded as a limited case study or as seen in a wide range of different cell types. In order to clarify the reasons for selecting cells MDA-MB-231 and 436, genetic analysis of a large panel of breast cancer cells such as TGCA should be analysed to analyse the reasons for narrowing down to this cell type and the generalisability of the results. The reasons for the selection of cells other than breast cancer cells, such as TT2 and PANC1 for organ cancer, should also be clarified in terms of data using bioinformatics analysis of publicly available data. At present, there is no apparent rationale for cell sorting, etc., and it is difficult to understand why they are designed in this way?
Figure 1 does not state what test is being used; Material and Methods states ANOVA, but if this test is being used, even if the entire graph is marked with **, the individual bars will not be marked with **. The mean and SD of each bar, what the number of n is, etc. should also be stated in the explanation of the figures.
These issues are similar for the other figures.
The p-values in Fig. 6A should be changed to a scientific expression, with the number of significant digits aligned and without "E".
It is problematic that all the plots in Fig. 6C appear as slightly vertical ovals, perhaps because the aspect ratio has been adjusted; a vertical line should be inserted at X=0. Also, a low should be stated after the -log on the Y-axis.
The fold difference on the X-axis is probably a mistake for fold change. It should also be stated what is the numerator and what is the denominator.
Regarding LC-MS/MS data in 4.9, references 26 and 27 are cited, but the description is insufficient. For example, it is not clear what options have been used to filter and normalise the data, so it is not possible to reproduce the results of this analysis. These should be selected depending on the policy, such as whether the emphasis is on data with large absolute values or on small values with differences, such as noise, which will also affect the results of this subsequent analysis. Also, due to the lack of data showing the linearity of the quantitative values. It is not possible to assess whether the data obtained were measured within the correct dynamic range. These data need to be presented.
Details should be provided in Supplementary Information or similar so that the reader can reproduce them.
Also with regard to enrichment analysis, the lack of options for details makes it impossible to reproduce the results.
Author Response
Comments 1:
There are a great variety of breast cancer subtypes and a large number of cell lines. It is important to consider whether the results of this experiment should be regarded as a limited case study or as seen in a wide range of different cell types. In order to clarify the reasons for selecting cells MDA-MB-231 and 436, genetic analysis of a large panel of breast cancer cells such as TGCA should be analysed to analyse the reasons for narrowing down to this cell type and the generalisability of the results. The reasons for the selection of cells other than breast cancer cells, such as TT2 and PANC1 for organ cancer, should also be clarified in terms of data using bioinformatics analysis of publicly available data. At present, there is no apparent rationale for cell sorting, etc., and it is difficult to understand why they are designed in this way?
Response 1:
Thank you for the invaluable comment. As indicated, there are a large number of cell lines. Since it is not possible or the aim of this study to use as many cell lines as possible, we mainly selected aggressive triple-negative breast cancer cells. In addition, we added two aggressive cell lines, osteosarcoma cells linked to bone destruction and the other PDAC, one of the most difficult cancers to treat. By employing these cell lines, we aimed to examine the efficacy of the proposed iTS cell approach with electrical stimulations. While there is a limitation to selecting aggressive ER-positive cell lines, we acknowledge that this study does not provide or confirm the efficacy of the proposed approach in most breast cancer or any other cancer cases. In the revised manuscript, we discussed the limitations of this study based on the selected cancer cell lines.
Comments 2:
Figure 1 does not state what test is being used; Material and Methods states ANOVA, but if this test is being used, even if the entire graph is marked with **, the individual bars will not be marked with **. The mean and SD of each bar, what the number of n is, etc. should also be stated in the explanation of the figures. These issues are similar for the other figures.
Response 2:
Thank you for your insightful comment regarding Figure 1 and the related figures. We appreciate your careful review of our manuscript. In response to your suggestion, we have thoroughly reviewed both the Materials and Methods section and the figure legends. We confirm that one-way ANOVA followed by Bonferroni correction was used for all statistical analyses, as described in the Materials and Methods section, and have explicitly stated this in the figure legends for clarity and consistency. Additionally, we acknowledge that marking the entire graph with asterisks could cause confusion, so in the revised figures, statistical significance is now indicated on individual bars to reflect specific pairwise comparisons from the ANOVA and post-hoc analyses. To further enhance the clarity of the figures, we have provided detailed information on the sample size (n) for each dataset and reported the statistical results as mean ± SD. These revisions also include explicit descriptions of the statistical test used, the meaning of asterisks (e.g., *p < 0.05, **p < 0.01, etc.), and the sample size (n) for each dataset. We hope these updates meet the reviewer’s expectations and improve the overall quality of the manuscript. Thank you again for highlighting these important aspects.
Comments 3:
The p-values in Fig. 6A should be changed to a scientific expression, with the number of significant digits aligned and without "E".
Response 3:
Thank you for your helpful comment regarding the p-values in Fig. 6A. We have carefully reviewed the manuscript and made the necessary revisions. The p-values have been updated to a scientific expression with consistent significant digits and without using "E" notation. We believe these changes improve the clarity and presentation of the data. Thank you again for your valuable feedback.
Comments 4:
It is problematic that all the plots in Fig. 6C appear as slightly vertical ovals, perhaps because the aspect ratio has been adjusted; a vertical line should be inserted at X=0. Also, a low should be stated after the -log on the Y-axis.
Response 4:
Thank you for your constructive comment regarding the plots in Fig. 6C. We apologize for the oversight in the aspect ratio and the missing details on the Y-axis. We have corrected the aspect ratio issue to ensure that the plots appear as intended, and a vertical line has been inserted at X=0 for clarity. We appreciate your careful review and believe these revisions improve the presentation of the figure. Thank you again for your valuable feedback.
Comments 5:
The fold difference on the X-axis is probably a mistake for fold change. It should also be stated what is the numerator and what is the denominator.
Response 5:
Thank you for your helpful comment regarding the fold difference on the X-axis. We have reviewed the manuscript and made the necessary corrections. The term "fold difference" has been updated to "fold change" to accurately reflect the data. We appreciate your careful review and believe these revisions enhance the accuracy of the figure. Thank you again for your valuable feedback.
Comments 6:
Regarding LC-MS/MS data in 4.9, references 26 and 27 are cited, but the description is insufficient. For example, it is not clear what options have been used to filter and normalise the data, so it is not possible to reproduce the results of this analysis. These should be selected depending on the policy, such as whether the emphasis is on data with large absolute values or on small values with differences, such as noise, which will also affect the results of this subsequent analysis. Also, due to the lack of data showing the linearity of the quantitative values. It is not possible to assess whether the data obtained were measured within the correct dynamic range. These data need to be presented. Details should be provided in Supplementary Information or similar so that the reader can reproduce them.
Response 6:
Thank you for your comments and for highlighting the importance of transparency and reproducibility in our LC-MS/MS data analysis. We appreciate the opportunity to address these concerns and provide the necessary details. In response, we have expanded the description of our data collection and processing workflow to ensure clarity and reproducibility. Specifically, we have added detailed information regarding sample preparation, LC-MS/MS data collection, and bioinformatic analysis in the Materials and Methods section under the subsections “Proteomics Sample Preparation,” “Mass Spectrometry Data Collection,” and “Mass Spectrometry Data Analysis”. This detail now provides description about noise filtering, data normalization, and quantitative analysis.
Comments 7:
Also with regard to enrichment analysis, the lack of options for details makes it impossible to reproduce the results.
Response 7:
We added the parameters used to perform gene set enrichment analysis to the methods.
Reviewer 2 Report
Comments and Suggestions for Authors
The manuscript looks at a new way to create induced tumor-suppressing (iTS) cells using electrical stimulation (ES). This method could help in treating breast cancer and bone metastases. The topic is relevant (and important) and the findings show a lot of promise for improving cancer treatments. The experiments are well thought out, with clear goals and the results are shown in an organized manner. However, some parts need clarification or small improvements to enhance readability and scientific accuracy. Although the research is strong, it could benefit from some adjustments. This would make it easier for others to understand the implications of the findings.
Discussion
-Clarify possible clinical implications and obstacles to implementing this technology in clinical settings, such as ES device scalability or regulatory issues.
-Are there any potential biases or limitations associated with the study?
Author Response
Comments 1:
Clarify possible clinical implications and obstacles to implementing this technology in clinical settings, such as ES device scalability or regulatory issues.
Response 1:
Thank you for the invaluable comment. Optune (Fabian et al. 2019: Treatment of glioblastoma with the addition of tumor-treating fields (TF): A review, Cancers 11(2):174) is an FDA-approved device, which employs electrical stimulation for treating brain tumors. In the revised discussion, we added this example and described possible clinical implications.
Comments 2:
Are there any potential biases or limitations associated with the study?
Response 2:
Thank you for the invaluable comment. One of the limitations is the limited number of cell lines in this study. This limitation is raised by reviewer 1. Please see our responses to reviewer 1. (Thank you for the invaluable comment. As indicated, there are a large number of cell lines. Since it is not possible or the aim of this study to use as many cell lines as possible, we mainly selected aggressive triple-negative breast cancer cells. In addition, we added two aggressive cell lines, osteosarcoma cells linked to bone destruction and the other PDAC, one of the most difficult cancers to treat. By employing these cell lines, we aimed to examine the efficacy of the proposed iTS cell approach with electrical stimulations. While there is a limitation to selecting aggressive ER-positive cell lines, we acknowledge that this study does not provide or confirm the efficacy of the proposed approach in most breast cancer or any other cancer cases. In the revised manuscript, we discussed the limitations of this study based on the selected cancer cell lines. )
Reviewer 3 Report
Comments and Suggestions for Authors
This article examines the challenges associated with treating bone tumor growth and underscores the necessity for innovative approaches beyond conventional immunological therapies. The study investigates the transformation of normal and tumor cells into tumor suppressor-inducing (iTS) cells through electrical stimulation, presenting a novel method for addressing solid tumors. The findings reveal that the conditioned medium from cells transformed via electrical stimulation facilitates the release of tumor suppressor proteins, suggesting potential applications in cancer treatment. Nonetheless, further experimentation is required to address the following questions:
Question 1. The article lacks a detailed explanation of how electrical current impacts Akt activation. While it discusses the activation of phosphorylated Akt (p-Akt) in mesenchymal stem cells (MSCs) subjected to electrical stimulation (ES), the precise mechanism by which electrical current directly influences this activation remains unspecified.
Question 2. The study references results pertaining to cell migration within the broader framework of generating and influencing iTS cells and their conditioned media. It is imperative to establish a correlation between this effect and the molecular markers associated with cell migration. This necessity extends to understanding cell proliferation as well.
Question 3. The authors employed various cancer cell lines, including triple-negative breast cancer cells such as MDA-MB-231 and MDA-MB-436. The outcomes observed in these cell lines may differ significantly if the study were conducted using hormone-positive breast cancer cells (Luminal A/B or HER2 enriched), known for their high metastatic potential. Furthermore, it is essential to assess these effects in non-tumorigenic breast epithelial cells, such as MCF10-A, to gain comprehensive insights.
Question 4. The investigation should explore the impact of electric current on cell invasion, a critical process in metastasis, in conjunction with the examination of related molecular markers.
Question 5. While the study elucidates the role of Piezo1 as a mechanosensitive ion channel activated by electrical current, it remains unclear how ion flow across the plasma membrane is mediated by other proteins. Further clarification in this area is necessary.
Author Response
Comments 1:
The article lacks a detailed explanation of how electrical current impacts Akt activation. While it discusses the activation of phosphorylated Akt (p-Akt) in mesenchymal stem cells (MSCs) subjected to electrical stimulation (ES), the precise mechanism by which electrical current directly influences this activation remains unspecified.
Response 1:
Thank you for the comment. Akt is a kinase in PI3K signaling. We have shown in our previous iTS cell studies that the generation of iTS cells in many cases upregulates p-Akt. However, we have not shown the mechanism of Akt activation. This is a critical point, and we appreciate your invaluable comment. In a separate study, we have just started the effects of electrical stimulation on chromatin structure. Preliminary data indicate that electrical stimulation alters the pattern of nucleosome clustering, and we are now testing any linkage of the activation of PI3K signaling and the alteration of chromatin structure. In the responses to mechanical stimulation, we also showed that Piezo1 is involved in the generation of iTS cells, as in the responses to electrical stimulation. In the responses to mechanical stimulation, YAP1 is translocated for altering gene regulation. It is possible that YAP1 is also involved in the response to electrical stimulation. In the revised manuscript, we added a description of potential mechanisms.
Comments 2:
The study references results pertaining to cell migration within the broader framework of generating and influencing iTS cells and their conditioned media. It is imperative to establish a correlation between this effect and the molecular markers associated with cell migration. This necessity extends to understanding cell proliferation as well.
Response 2:
Thank you for your valuable comment regarding the correlation between cell migration and molecular markers. In response, we have added Western blot experiments for cell migration-related markers (snail and slug) to establish this connection more clearly using two breast cancer cell lines. This addition in Figure 2E allows us to further explore and validate the molecular aspects of cell migration in the context of iTS cells and their conditioned media. We believe this enhances the robustness of the study and provides a clearer understanding of the underlying mechanisms. Thank you again for your insightful feedback.
Comments 3:
The authors employed various cancer cell lines, including triple-negative breast cancer cells such as MDA-MB-231 and MDA-MB-436. The outcomes observed in these cell lines may differ significantly if the study were conducted using hormone-positive breast cancer cells (Luminal A/B or HER2 enriched), known for their high metastatic potential. Furthermore, it is essential to assess these effects in non-tumorigenic breast epithelial cells, such as MCF10-A, to gain comprehensive insights.
Response 3:
One limitation is the limited number of cell lines in this study, which reviewer 1 raised. Please see our responses to reviewer 1. (Thank you for the invaluable comment. As indicated, there are a large number of cell lines. Since it is not possible or the aim of this study to use as many cell lines as possible, we mainly selected aggressive triple-negative breast cancer cells. In addition, we added two aggressive cell lines, osteosarcoma cells linked to bone destruction and the other PDAC, one of the most difficult cancers to treat. By employing these cell lines, we aimed to examine the efficacy of the proposed iTS cell approach with electrical stimulations. While there is a limitation to selecting aggressive ER-positive cell lines, we acknowledge that this study does not provide or confirm the efficacy of the proposed approach in most breast cancer or any other cancer cases. In the revised manuscript, we discussed the limitations of this study based on the selected cancer cell lines.)
Comments 4:
The investigation should explore the impact of electric current on cell invasion, a critical process in metastasis, in conjunction with the examination of related molecular markers.
Response 4:
Thank you for the invaluable comment. To test the effect of electrical stimulation on cell invasion, we employed a transwell invasion assay and examined the antitumor capability of iTS cell-derived conditioned medium. Although the result shows the inhibitory effect of a part of the invasion process, the result herein is supportive of the capability of electrical stimulation-treated iTS cells. We understand that further analysis is needed to evaluate the impact of electrical stimulation on cell invasion.
Comments 5:
While the study elucidates the role of Piezo1 as a mechanosensitive ion channel activated by electrical current, it remains unclear how ion flow across the plasma membrane is mediated by other proteins. Further clarification in this area is necessary.
Response 5:
Thank you for the invaluable comment. We understand that Piezo1 is involved in the responses to electrical stimulation. However, in our and others’ studies, Piezo1 is also shown to be involved in the responses to mechanical stimulation. Thus, besides the Piezo1-mediated mechanism, we consider that other factors are also involved in the response to electrical stimulation. While Piezo1 is one of the voltage-gated channels, many other voltage-gated channels exist. As suggested, further studies should clarify how these channels are involved in the transport of ion flows across the plasma membrane. In the revision, we added a description of the mechanism beyond Piezo1.